# The Incidence and Nature of Claims against Dentists Related to Periodontal Treatment in Israel during the Years 2005–2019

**DOI:** 10.3390/ijerph18084153

**Published:** 2021-04-14

**Authors:** Dima Nassar, Nirit Tagger-Green, Haim Tal, Carlos Nemcovsky, Eitan Mijiritsky, Ilan Beitlitum, Eitan Barnea, Roni Kolerman

**Affiliations:** 1The Maurice and Gabriela Goldschleger School of Dental Medicine, Tel-Aviv University, Tel-Aviv 6997801, Israel; 2General Practitioner, Private Clinic, Tel-Aviv 6100000, Israel; 3Department of Periodontology and Implant Dentistry, The Maurice and Gabriela Goldschleger School of Dental Medicine, Tel-Aviv University, Tel-Aviv 6997801, Israel; drnirit@gmail.com (N.T.-G.); talhaim@tauex.tau.ac.il (H.T.); carlos@tauex.tau.ac.il (C.N.); beilan1612@gmail.com (I.B.); kolerman@netvision.net.il (R.K.); 4Department of Oral Rehabilitation, The Maurice and Gabriela Goldschleger School of Dental Medicine, Tel-Aviv University, Tel-Aviv 6997801, Israel; mijiritsky@bezeqint.net; 5Department of Otolaryngology Head and Neck Surgery and Maxillofacial Surgery, Tel-Aviv Sourasky Medical Center, Sackler School of Medicine, Tel Aviv University, Tel-Aviv 6997801, Israel; 6Prosthodontist, Private Practice, Tel-Aviv 6100000, Israel; drbarnea@gmail.com

**Keywords:** periodontal malpractice, claims, adverse events

## Abstract

Background: In recent years, worldwide dental malpractice claims have dramatically increased. The purpose of the present study is to analyze periodontal therapy related claims in Israel that led to legal decisions. Methods: This retrospective cohort study includes malpractice claims against dental practitioners related to periodontology between 2005 and 2019. Only closed cases where a final decision was made were included. The chi-square test or Fisher exact test for categorical variables were performed and a *p* value of <0.05 was considered statistically significant. Results: During the study period there were 508 legal claims related to periodontal disease. Most plaintiffs were women (63.4%), and 71.3% of the patients were >35 years old. Most claims (82.8%) were settled out of court and ended in compromise. Claims concerning the treatment of periodontal disease by periodontists accounted only for 4.5% (23/508) of the cases while 95.5% (485/508) of the claims were for complications secondary to another treatment. Prosthodontic treatment was involved with the highest number 54.5%, followed by dental implants 17.7%, and endodontics 11.6%. The allegations were related to pain and distress (84.8%), aggravation of existent periodontal disease (83.3%), tooth loss (78.1%), and violation of autonomy (47%). Conclusions: The main cause for lawsuits was related to aggravation of periodontal disease during prosthetic or implant therapy and related to suspected faulty or inexistent preoperative diagnosis and planning. Practical implications: Periodontal consultation before dental treatment may reduce malpractice risks, adverse events, and un-necessary changes of treatment plans.

## 1. Introduction

Malpractice claims related to dental treatment are the most frequent among the medical profession [1,2]. Moreover, the number of legal proceedings involving dental professional liability has progressively increased in Israel as in other countries [3,4,5,6]. In Israel, the probability of a dentist to be sued is 12% [1]. In Australia, over 10.5% of the legal claims in the medical field were against dental practitioners, and almost 16% of the dentists were the subject of at least one legal suit [2]. These facts could be related to patients’ perception that all unsuccessful dental treatment outcomes are due to professional misconduct [3,5]. Additional factors are the high costs of dental care combined with, sometimes, unrealistic patients’ expectations. This has been shown in an Italian study [7] and is also relevant for Israel where most dental treatment costs are privately covered.

Since periodontal disease is very common, affecting 42.2% of the US population with 7.8% experiencing severe forms [8], clinical and radiographic screening for signs and symptoms of the disease are mandatory [9,10]. There are no updated data concerning the prevalence of periodontal disease in Israel [11]. The only survey in Israel concerning periodontal treatment needs was performed in 1989 [12]. The study revealed that 29.6% of the medical personnel had periodontal pockets deeper than 5 mm, and on average 0.61 of the sextants had deep pockets.

Data on claims related to the diagnosis, management, and outcomes of periodontal disease are scarce, although this information is highly relevant for dental practitioners revealing the most frequent errors, as perceived by their patients. Additionally, dental professionals can reconsider their own practices and adopt improved risk-prevention procedures by becoming more aware of medico-legal risks and the dentist–patient interrelationship [13].

The aim of the present study is to retrospectively analyze the legal claims directly and/or indirectly related to periodontal treatment in Israel between the years 2005 and 2019 based on the computerized database of the “Medical Consultant International” (MCI) company (Subsidiary of Madanes group), insuring almost 95% of the dental practitioners in Israel [4].

## 2. Materials and Methods

All legal claims, in which a decision was made (closed claims), related to diagnosis, management, and outcomes of periodontal disease registered in the computerized database of MCI from January 2005 until December 2019 were evaluated.

The study was approved by the Tel-Aviv University Ethical Committee.

Inclusion criteria: Claims related to diagnosis, management, and outcomes of periodontal disease according to the MCI registry.Files including full relevant data: gender and age of the patient, date of the complaint, treatment setting (private, public, or corporate clinic), a detailed description of the adverse event, nature of allegations, clinical outcome of injury, outcome of claim, and payment amounts.

Data were classified according to: (A) the main reason for the claims, (B) the clinical setting of treatment, (C) the litigation outcome.

All the data used by the researchers were anonymous, comprising only the necessary information to avoid duplication, date, location, and detailed description of the complaint. 

The following variables were analyzed:Demographics: data were compared between female and male patients and between age groups (≤/>35 years at presentation).The main reasons for the claims: this was further divided into subgroups:
(a)The management of periodontal disease by periodontists: this included, injuries during periodontal therapy including neural damage, tooth/teeth loss, root resorption, patients’ disappointment of the esthetic outcomes (recession aggravation, enlargement of interdental embrasure), thermal hypersensitivity, increased tooth/teeth mobility, and needs for re-treatment.(b)Secondary periodontal complications occurring during non-periodontal treatment, such as prosthetic rehabilitation, implant therapy, tooth extraction, orthodontic, restorative, endodontic, pedodontic, or any other type of dental/oral treatment. This parameter was further classified according to the exact reason for the claim: delayed or false diagnosis, delay in treatment, needs for further treatment due to lack of disease, diagnosis and complications following the treatment (neural damage, esthetic damage, distress and pain, tooth/teeth loss and needs for additional treatment).(c)Documentation/information: lack of relevant information regarding the treatment performed and possible complications and lack of a detailed informed consent.The clinical setting where treatment was provided:
(a)private dental practices (PDC)(b)public dental practices (PDP)(c)corporations (private dental clinics registered under one juristic entity).Litigation outcome:
(a)compromise out of court between insurance company and claimant(b)mediation—in court compromise(c)rejection—rejection of the malpractice claim by the insurance company without any further litigation(d)cancellation—claim cancellation by the court(e)court adjudicated—in court verdict(f)insurance rejection—rejection of the claim as the treatment was not performed during insurance policy period(g)accepted arbitration at the Israeli Dental Association Court(h)file closed without compensation—the court litigation decision was that the claimant is not entitled to any compensation.

Two researchers (DN, RK) independently screened and interpreted all files applying the same criteria. In cases of disagreement, data were discussed together with another author (NTG) until an agreement reached.

## 3. Statistics

The groups were compared using the chi-square test or the Fisher exact test for categorical variables. A *p* value of <0.05 was considered statistically significant. The SPSS software was used for all statistical analyses (IBM SPSS Statistics for Windows, version 24, IBM Corp., Armonk, NY, USA, 2016).

Due to the uneven numbers of the genders, the statistical evaluation was performed within groups considering the relative percentage of patients of both genders in each group.

## 4. Results

During the study period, 644 claims were litigated against dentists related to periodontology. Overall, 508 cases were ended by a decision either by the insurance company or by the courts. The cases that remained open and did not receive a decision were excluded from the statistical analysis. Demographic data of the patients, treatment setting, and litigation results are shown in Table 1.

Table 1: The numbers and percentages (%) of claims by patients’ gender and age, treatment setting, and litigation results in a cohort of 508 patients. Women made up 63.4% of the cohort (322/508). The average age of the claimants was 44.62 ± 14.83 years (range 11–84, median—45.00) and 71.3% (362/508) of the patients were more than 35 years old (Table 1).

The majority of the litigation processes ended with a compromise. The claim was seldom rejected. Most claims were against dentists working in private clinics. Public dental clinics and corporations were combined as “Public Sector” for further analysis and discussion.

### 4.1. The Purpose of the Claim as a Function of the Primary Treatment Performed

Only 23 claims (4.5%) were directly related to the management of primary periodontal disease, while 95.5% (485/508) concerned periodontal complications secondary to another treatment (Figure 1).

Most of them related to prosthodontic treatment (277 claims—54.5%), followed by dental implants (90 claims—17.7%), and endodontics (59 claims—11.6%). (Figure 1) Lawsuits concerning the remaining procedures in a descending order were orthodontics (35 claims—6.9%), tooth extraction (15 claims—3.0%), oral medicine/pathology (8 claims—1.6%), and pedodontics (1 claim—0.2%) (Figure 1).

### 4.2. Allegation Nature by General Category

Most claims were related to other discipline treatments (prosthetic rehabilitation, implants, orthodontic treatment, etc.) performed while ignoring or delaying periodontal therapy leading to aggravation of periodontal disease, treatment delay, over-treatment, and delay of diagnosis (Table 2).

Periodontal claims against periodontists were very sparse accounting only for 4.5% of the cases (Table 2).

It should be noted that a relatively high number of claims accounted for violation of autonomy, meaning that patients were not precisely informed about the treatment plan, its possible complications and side effects, and did not sign an informed consent form.

Reasons for the claims: in most cases the operators were sued for more than one reason.

The most frequent claims related to the clinical outcomes of neglected periodontal disease treatment were pain and distress; tooth loss/damage caused mainly due to prosthetic or implant therapy. Endodontic treatment was the third common reason for tooth loss. Claims for tooth loss due to periodontal therapy were uncommon. Other reasons for claims included needs for further surgical treatment, disappointment of the esthetic outcomes, and neural damage (Table 3).

Claims by age: there were statistically more claims in the younger group (≤35) due to delay in diagnosis and/or treatment (*p* < 0.001, *p* = 0.004, respectively). The oldest group (>35) sued significantly more because of changes in the treatment plan (*p* = 0.023), over-treatment (*p* = 0.028), periodontal disease aggravation (*p* = 0.002), and documentation/information related issues.

Claims by treatment setting: dentists in the public offices were sued significantly more for delay in diagnosis (*p* = 0.009) and false diagnosis (*p* = 0.029). In contrast, the private offices were sued significantly more for neural damage (*p* = 0.003), violation of autonomy (*p* = 0.032), distress and pain (*p* < 0.001), and aggravation of periodontal disease (*p* = 0.017).

Claims over the years: in the recent years (2012–2019), patients claimed significantly more for neural damage (*p* = 0.001), violation of autonomy (*p* < 0.001), over-treatment (*p* = 0.005), distress and pain (*p* = 0.018). There were significantly more claims for periodontal malpractice in recent (2012–2018) than in past years (2005–2011) (*p* = 0.032) (Figure 2).

### 4.3. Claims According to the Treatment Performed

Women sued statistically more for periodontal disease aggravation caused due to prosthetic treatment (61.9% vs. 44.8%), and more men sued for implant (24.5% vs. 14.3%) and endodontic reasons (17.9% vs. 8.3%).

Comparing private and corporate offices revealed that patients in the corporate offices sued statistically more (*p* = 0.001) for implant therapy (25.0%) than patients in private offices (9.7%).

Patients aged under 35 years sued significantly more than the older for orthodontic therapy (21.2% vs. 1.1%, respectively) and endodontic treatment (18.1% vs. 9.3%, respectively). While the older patients sued more for prosthetic (58.0% vs. 45.9%, respectively) and implant therapy (22.4% vs. 6.2%, respectively).

## 5. Discussion

The study population included 508 claims related to periodontal therapy, in a period of over 15 years (2005–2019). In comparison, the number of settled claims in implant dentistry in Israel during 2005–2015 was much higher, accounting for 709 cases [13], a similar finding [5] and an even lower number of periodontal therapy-related claims [14,15,16,17] have been reported. According to a previous study [13], the mean time lag between time of event to conclusion of a claim was 44.89 ± 28.7 months (range 1 to 233), with no change over the years, thus an escalation in number of claims related to periodontology during the 2016–2019 period can be expected. While in certain countries, including Finland [3], Iran [5], and the UK [18], a rise in the number of claims in the fields of implantology and prosthodontic has been recorded, in others [14] the trend is opposite.

Compared with previous reports [3,6,19,20], in the present study female patients sued almost twice as often as males (63.4% vs. 36.6%), the suggested reasons could be a greater interest in dental health and use of services [21]. The meaning of this finding should not be that female patients tend to litigate more than male patients, but that women seek treatment more often, and therefore, represent the majority of the treated patients.

Similar to previous data [6], in the present report, only 8.9% of the claims did not received any compensation; however, others have reported a higher percentage of claim rejection [16,17,21,22].

A discussion on the reasons for the relatively high proportion of patient compensation cases is beyond the scope of this study, but it might be associated to the policy of the insurance company [23].

A large percentage of claims (54.5%) were related to prosthetic treatment, followed by dental implants (17.7%). A frequent complain was implant placement in active periodontal disease patients, which in turn, may lead to extractions during/after implant placement and consequently to high unplanned costs [24,25,26].

Endodontics was also related to periodontal claims (11.6%). Trauma, root resorptions, perforations, cracks, and dental malformations play an important role in the development and progression of lesions resulting from combined endodontic periodontal diseases or may manifest as periodontitis [17,27,28].

Orthodontics was involved in 6.9% of the claims. When properly performed, an orthodontic treatment may improve the tooth position, creating access for oral hygiene, and altering occlusal factors; however, if performed in patients with active periodontal disease, it usually leads to additional periodontal attachment loss [29].

Oral medicine/pathology involves only 1.6% of the claims. In the present study, dentists were sued for not diagnosing life-threatening lesions, such as squamous cell carcinomas, ameloblastoma, osteomyelitis, and other tumors [30,31].

Public offices were sued more than the private ones due to delay in diagnosis and false diagnosis. In contrast, private offices were sued significantly more for neural damage, violation of autonomy, distress and pain, and aggravation of periodontal disease. Neural damage complaints, usually related to implant installation, has been increasing during the last few years, a statistic that can be explained by the fact that in Israel, as in most European countries, these treatments are equally performed by general dental practitioners as well as specialists [32].

There were a few significant differences between the younger (up to 35 years) and the older group (more than 35); this age split was made due to the traditional classification of early onset versus adult periodontitis characteristic to these age groups [33]. Based on the results, we believe that in addition to objective reasons there might be an age-related, different attitude regarding filing a claim. The younger filed more claims due to delay in diagnosis and/or treatment. The oldest group sued significantly more due to over-treatment, periodontal disease aggravation, changes in treatment plans, and documentation-information issues. Claims associated with orthodontic treatment were, as could be expected, significantly more frequent in the younger patients. This may be due to the high proportions of young people appealing for orthodontic treatment and that periodontal disease is less frequent in these patients and therefore could be overlooked. As could be expected, sues related to prosthetic and implant therapy were more frequent in the older patients [34].

In the recent years (>2011), patients claimed significantly more for neural damage, violation of autonomy, over-treatment, distress, and pain. Lately, more extensive and complex treatments are performed by general practitioners resulting in a rise in the number of malpractice claims.


Strengths and limitations:


The MCI database used in this study covers the entire country, as almost 95% of the dental practitioners in Israel were professionally insured by this company during the 14-year study period. This is a major strength of our cohort.

However, the study has several limitations. First, the relevance of the results on the subgroups divided by age, gender, etc. is limited, as the division of the total number of patients undergoing dental and periodontal treatment according to these subgroups is unknown. For example, although most claimants in our study were women, we cannot conclude that it is riskier to treat a female patient from a malpractice perspective. Instead, it may reflect the fact that women are the majority of treated patients. A second limitation is that, although we described the total number of claims filed during the study period, we analyzed only cases settled with an established decision. Another limitation is that we could not get further demographic data concerning the claimants—such as level of education and income. The only demographic data we got were gender and age.

There is some difficulty comparing between countries due to different demographics, but we intended to highlight a special issue regarding periodontal malpractice claims. This important issue was never investigated, although it is related to every dental treatment.

Finally, due to the objection of the insurance company, data on the compensation payments were not included.

## 6. Conclusions

Based on our findings, some practical recommendations are proposed for reducing malpractice events and claims:Practitioners should dedicate a significant part of the total treatment time to preoperative diagnosis and planning. Treatment plans should be outlined in writing and be detailed. The dentist must inform the patient of the risks and benefits of the proposed treatment, the consequences of declining treatment, and the available treatment options. Alternative treatment plans should be discussed with the patient and documented. It is inevitable that, if a dentist carries out dental treatment without the consent of the patient, he/she will face the consequences.It is preferable to develop a full cause-related therapy and maintenance program. A skilled dentist is supposed to carry out an optimal treatment plan with minimal changes in most clinical situations (and potential changes must be acknowledged).All documents should be signed and retained, including treatment-specific informed consent (sinus augmentation, bone augmentation, immediate placement, etc.) and financial agreement to undergo the treatment plan.The main causes for lawsuits are actual body injury and major disappointment with the results of the therapy. The combination of proper operative skills and a good doctor–patient relationship will reduce the number of legal claims.Periodontal consultation before dental treatment may reduce malpractice risks, adverse events, and un-necessary changes of the treatment plan. The most common issues of claims were related to primary “non-periodontal patients” meaning that dental treatment was supplied while ignoring the under/delayed treating of the periodontal disease. Once established, it should be clearly presented to the patient, including the risks and benefits, alternative treatment options, and possible complications.

Litigation and malpractice claims are a part of everyday life in medical/dental practice. The judicious clinician must be as diligent in risk-reduction management and strategies as he/she is in practicing excellent dentistry. Good patient communication, relationship, and excellent documentation are the keys to minimizing and possibly eliminating future lawsuits.

## Figures and Tables

**Figure 1 ijerph-18-04153-f001:**
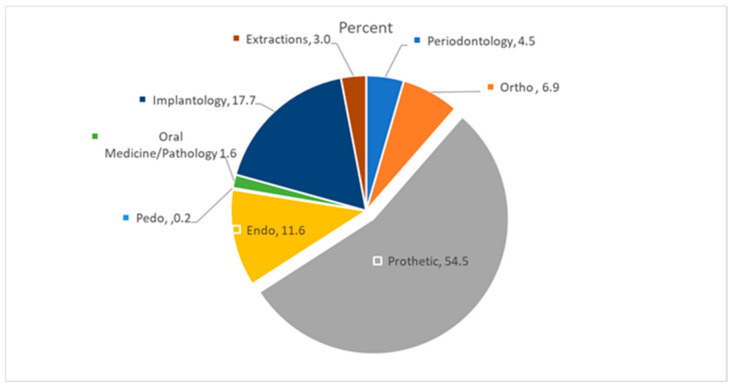
The purpose of the claim as a function of the primary treatment performed.

**Figure 2 ijerph-18-04153-f002:**
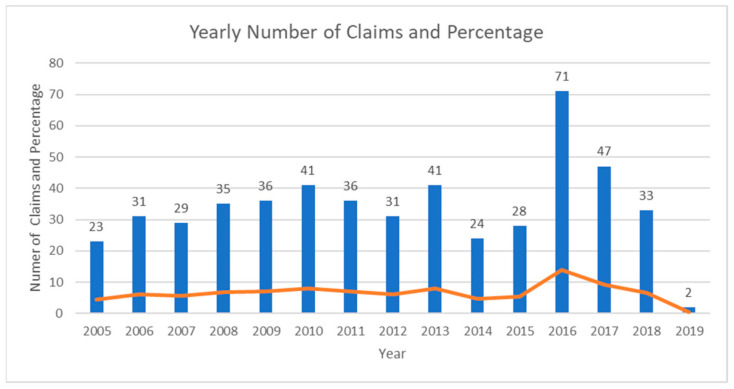
Yearly number of claims and percentages.

**Table 1 ijerph-18-04153-t001:** Patient demographic characteristics.

	Variable	*n*	%
Gender	Male	186	36.6%
	Female	322	63.4%
Age	>35	362	71.3%
	≤35	146	28.7%
Sector	Private	276	54.3%
	Public	198	39.0%
	Corporates	34	6.7%
Litigation status	Compensation by compromise	421	82.8%
	Compensation by court mediation	37	7.3%
	Court adjudicated compensation	5	1.0%
	Rejection	17	3.4%
	Cancellation	27	5.3%
	No compensation	1	0.2%
Total		508	

**Table 2 ijerph-18-04153-t002:** Allegation nature by general category—periodontal.

Non-Periodontal Treatment												
*n*, (% of Total)	Gender, *n* (%)			Age (Years), *n* (%)			Treatment Settings, *n* (%)			Year of Claim, *n* (%)		
	F	M	ρ	≤35	>35	ρ	Private	Public	ρ	≤2011	>2011	*p*
Periodontal deterioration in another treatment 485 (95.5)	307 (95.3)	178 (95.7)	1.000	138 (94.5)	347 (95.9)	0.488	265 (96.0)	220 (94.8)	0.530	221 (95.7)	264 (95.3)	0.879
Aggravation of periodontitis 423 (83.3)	148 (79.6)	275 (85.4)	0.09	109 (74.7)	314 (86.7)	0.002	240 (87.0)	183 (78.9)	0.017	187 (81.0)	236 (85.2)	0.233
Over-treatment 90 (15.1)	35 (18.8)	55 (17.1)	0.621	17 (11.6)	73 (20.2)	0.028	46 (16.7)	44 (19)	0.56	32 (13.9)	58 (20.9)	0.005
Delay of Treatment 106 (11.2)	62 (19.3)	44 (23.7)	0.240	43 (28.5)	63 (17.4)	0.004	51 (18.5)	55 (23.7)	0.149	42 (18.2)	64 (23.1)	0.189
Delay of Diagnosis 60 (8)	43 (13.4)	17 (9.1)	0.156	40 (27.4)	20 (5.5)	<0.001	23 (8.3)	37 (15.9)	0.009	25 (10.8)	35 (12.6)	0.528
False Diagnosis 27 (5.3)	18 (5.6)	9 (4.8)	0.716	16 (5.1%)	11 (5.6%)	0.490	9 (3.3)	18 (7.8)	0.029	14 (6.1)	13 (4.7)	0.054
Periodontal treatment by periodontists												
Periodontal surgery results 23 (4.5)	16 (5)	7 (3.8)	0.529	3 (2.1)	20 (5.5)	0.102	13 (4.7)	10 (4.3)	1.000	7 (3.0)	16 (5.8)	0.198
documentation/information												
Violation of autonomy 239 (47.0)	87 (46.8)	152 (47.2)	0.925	57 (39)	182 (50.3)	0.024	142 (51.4)	97 (41.8)	0.032	52 (22.5)	187 (67.5)	0.005
Change in Treatment plan 12 (1.8)	6 (1.9)	6 (3.2)	0.330	0 (0)	12 (3.3)	0.023	6 (2.2)	6 (2.6)	0.778	4 (1.7)	8 (2.9)	0.560

Statistically significant values (*p* < 0.05) are highlighted in red.

**Table 3 ijerph-18-04153-t003:** Clinical outcomes of neglect of periodontal disease treatment divided by gender, age, treatment setting, and years.

Clinical Outcomes of Allegations *n* (%)	Gender, *n* (%)			Age (Years), *n* (%)			Treatment Settings, *n* (%)			Year of Claim, *n* (%)		
	F	M	ρ	≤35	>35	ρ	Private	Public	ρ	≤2011	>2011	ρ
Distress and pain 431 (84.8%)	158 (84.9%)	273 (84.8%)	0.96	117 (80.1%)	314 (100%)	0.075	256 (92.8%)	175 (75.4%)	<0.001	186 (80.5%)	245 (88.4)	0.018
Tooth damage or loss 397 (78.1%)	154 (82.85)	243 (75.5%)	0.054	107 (73.35%)	290 (80.1%)	0.098	212 (76.8%)	185 (79.7%)	0.452	186 (80.5%)	211 (76.2%)	0.218
Re-Surgery 145 (28.5%)	61 (32.8%)	84 (26.1%)	0.107	39 (26.7%)	106 (29.3%)	0.589	84 (30.4%)	61 (26.3%)	0.325	64 (27.7%)	81 (29.2%)	0.767
Disappointment of the esthetic outcomes 114 (22.4%)	30 (16.1%)	84 (26.2%)	0.009	37 (25.5%)	77 (21.3%)	0.346	67 (24.4%)	47 (20.3%)	0.287	49 (21.3%)	65 (23.5%)	0.594
Neural Damage 21 (4.1%)	8 (4.3%)	13 (4%)	0.886	3 (2.1%)	18 (5%)	0.216	18 (6.5%)	3 (1.3%)	0.003	2 (0.9%)	19 (6.9%)	0.001

Statistically significant values (*p* < 0.05) are highlighted in red.

## Data Availability

Not applicable.

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
