# Peer review of "The Incidence and Nature of Claims against Dentists Related to Periodontal Treatment in Israel during the Years 2005–2019"

_ijerph, 2021, doi:10.3390/ijerph18084153_

Round 1
Reviewer 1 Report
The authors did well to address reviewer comments about the need for information about the periodontal and claims data in Israel and the need for more demographic information on the participants.
After the first sentence in the introduction (Lines 39-40 – "Malpractice claims related to dental treatment are the most frequent among the medical profession"), there should be a transition sentence before presenting information about Australia. Otherwise, the transition is a bit awkward especially when the paper title suggests the study is about periodontal treatment in Israel. Readers may ask, what is the connection between Israel and Australia? Before introducing information about Australia, the authors can mention that the information on malpractice claims is available in other countries such as Australia and also indicate the relevance of specially mentioning the country of Australia. Information is also mentioned for Italy. Once again, what is the connection between Israel, Australia, and Italy? Are those the only countries where this information is located? Do those countries have higher rates of malpractice claims and therefore, the authors wanted to use those countries as comparisons? It will be helpful to clarify this.
The expansion of the strengths and limitations and conclusion sections are good. For the conclusion, since the information is in numbered format, the authors should consider putting an introductory sentence before the numbered points and a concluding sentence after the numbered points.
Reviewer 2 Report
The manuscript in the present form for me is acceptable.
Round 2
Reviewer 1 Report
The authors have done a good job responding to the requested feedback. The paper is improved and appears suitable for publication.
This manuscript is a resubmission of an earlier submission. The following is a list of the peer review reports and author responses from that submission.
Round 1
Reviewer 1 Report
This study conducts a retrospective cohort study to assess incidence and nature of claims against dentists related to periodontal treatment in Israel between 2005-2019.
Authors aim to add to gaps in the literature by asserting that while information on diagnosis, management and outcomes of periodontal disease is highly relevant for dental practitioners, there is limited malpractice claims data on these topics.
In lines 49-51, authors state “Since periodontal disease is very common, affecting 42.2% of the US population with 7.8% experiencing severe forms, clinical and radiographic screening for signs and symptoms of the disease are mandatory.” This sentence seems misplaced since this paper is about periodontal treatment in Israel. Perhaps the authors can also provide these statistics as they apply to Israel if the information is available.
Moreover, much of the statistics provided in the introduction pertains to other countries such as Australia and Italy. The authors should supplement this information with more information about the statistics in Israel to bolster the introduction and establish a convincing argument on the need for this study.
Lines 80-81- Was any other demographic data collected other than age and gender? For instance, was information collected on level of education or income among participants? Are there other demographic variables that could have been assessed? If not, this could be discussed in the limitations section of the manuscript in terms of authors not being able to collect on this data.
The authors should also consider expanding their conclusion to include recommendations and future directions. What are the implications of the study results and how can these results be used to inform future research and practice?
Overall, this is an insightful and interesting paper on a pertinent topic. There are some clarifying questions that authors can attend to about the methodology, including data collection. The conclusion section should be expanded as well. Attending to these questions can provide clarity and help to improve the paper.
Reviewer 2 Report
Interesting study, but requiring some limitations to be discussed. The first thing that comes to the fore is that the authors only assessed court files, where each time the cases were assessed by different doctors who may have sometimes contradictory opinions. Looking at the problem's scale and the fact that the majority of cases ended with a compromise, this does not mean that the doctor made some omission or mistake. In that case, there would be no compromise.
I do not understand what is meant by incorrect periodontal treatment before prosthetic treatment. The basis of the decision of qualifying for prosthetic treatment is the completed conservative, endodontic, surgical, and possibly periodontal treatment.